# Influence of Atmosphere on Calibration of Radiation Thermometers

**DOI:** 10.3390/s21165509

**Published:** 2021-08-16

**Authors:** Vid Mlačnik, Igor Pušnik

**Affiliations:** Faculty of Electrical Engineering, University of Ljubljana, SI-1000 Ljubljana, Slovenia; igor.pusnik@fe.uni-lj.si

**Keywords:** transmissivity of air, radiation thermometer, calibration, measurement error, measurement uncertainty, moisture, humidity, infrared thermometer

## Abstract

Current process of calibrating radiation thermometers, including thermal imagers, relies on measurement comparison with the temperature of a black body at a set distance. Over time, errors have been detected in calibrations of some radiation thermometers, which were correlated with moisture levels. In this study, effects of atmospheric air on thermal transmission were evaluated by the means of simulations using best available resources of the corresponding datasets. Sources of spectral transmissivity of air were listed, and transmissivity data was obtained from the HITRAN molecular absorption database. Transmissivity data of molecular species was compiled for usual atmospheric composition, including naturally occurring isotopologs. Final influence of spectral transmissivity was evaluated for spectral sensitivities of radiation thermometers in use, and total transmissivity and expected errors were presented for variable humidity and measured temperature. Results reveal that spectral range of measurements greatly influences susceptibility of instruments to atmospheric interference. In particular, great influence on measurements is evident for the high-temperature radiation pyrometer in the spectral range of 2–2.7 µm, which is in use in our laboratory as a traceable reference for high-temperature calibrations. Regarding the calibration process, a requirement arose for matching the humidity parameters during the temperature reference transfer to the lower tiers in the chain of traceability. Narrowing of the permitted range of humidity during the calibration, monitoring, and listing of atmospheric parameters in calibration certificates is necessary, for at least this thermometer and possibly for other thermometers as well.

## 1. Introduction

Nonideal transmissivity of air is known in the field of radiation thermometry; however, the influence of this effect has not been evaluated and is usually neglected in the field. While the effect can be negligible for many radiation thermometer applications, evaluating this effect is of particular importance for improvement of practices in the fields where measurement uncertainty is important, including calibrations of reference thermometers, especially when ensuring traceability to the national standards.

The goal of this study is to analyze available resources of spectral data and calculate spectral transmissivity of air mixture for variable relative humidity, temperature, air pressure, and path length within expected usual atmospheric deviations. Ideally, a simple and fast MATLAB model of spectral transmissivity is to be made for evaluation of effect of thermal radiation transmission in the spectral range of 1 µm–18 µm, indicating the susceptibility of used laboratory equipment to atmospheric interference, and allowing repeated simulations for evaluation of measurement uncertainty in radiation thermometry.

### 1.1. Significance of the Research

In the Laboratory for Metrology and Quality of the Faculty of Electrical Engineering, University of Ljubljana, calibrations of radiation thermometers have been conducted for more than 20 years. The calibration procedure consists of a reference black body cavity at stabilized temperature, measured by the ITS-90 calibrated reference thermometer. The device in calibration is directed to the black body source from a set distance. Besides the contact thermometers, reference radiation thermometers are often used to measure the radiant temperature as well.

Between the consecutive calibrations of the reference radiation thermometer, drift has been detected, with the magnitude exceeding the total calibration uncertainty. Upon further investigation into causality, the drift has been found to correlate with different humidity levels during the calibration.

Usual calibration procedures of the calibration laboratory assume either total transmission of thermal radiation at short distances or fixed setting with negligible deviations. Regardless, humidity correction is always omitted, whereas the allowed parameter ranges, which are listed in the calibration certificate, are relatively wide. Atmospheric influences should be accounted for in the calibration process, either by matching calibration and target usage parameters, applying correction, and/or expanding calibration uncertainty.

### 1.2. Existing Works and Research

Earlier attempts of calculating thermal transmission were conducted by Elder and Strong [1] in 1953, where the total transmission was considered as the effect of continuous attenuation of particles and band absorption, also citing Beer’s law. Average influence of particles of water and the effect of pressure were later approximated for eight spectral ranges, from 0.7 µm to 14 µm. Calculation of overall transmission was subsequently reduced to eight estimations of the average radiance and transmissivity and thus limited resolution for calculations of transmission of thermal radiation, especially in the case of uneven spectral emissivity distribution of the radiation source.

Wyat et al. [2] calculated the spectral transmissivity of water vapor and carbon dioxide as a function of wavenumber, pressure, and path-times-length product, computed from vibrational and rotational constants. Included in the computed spectrum were the most strong vibrational transition lines, yielding sufficient resolution, however results were of limited accuracy when compared to measurements by Burch et al. [3].

An experimental concept has recently been proposed by Zhang et al. [4], where total transmissivity of a radiator–imager system is proposed to be calibrated against varying moisture levels in order to measure relative humidity of air with high response rate.

The first detailed computer models for public use were developed by the Air Force Geophysics Laboratory (AFGL). The Fortran-based Low-Resolution Transmission Model (LOWTRAN) [5] was developed with relatively low spectral resolution of 20 cm^−1^, with intention to simulate effects of the Earth’s atmosphere on transmission of radiation by the Sun, Earth, Moon, and horizontal observers in the lower atmosphere. The Moderate-Resolution Transmission Model (MODTRAN) [6] is the updated model of LOWTRAN, with the improved resolution of 2 cm^−1^. MODTRAN calculations were based on Voigt line shape, incorporating both the pressure-dependent Lorentz line shape and the temperature-dependent Doppler line shape, improving accuracy of modelling high-altitude atmospheric conditions. Continuation of these models also include Fast Atmospheric Signature Code (FASCODE), Line-By-Line Radiative Transfer Model (LBLRTM) and Combined Atmospheric Radiative Transfer Model (CART) [7]. Updates were focused on improving reliability and precision of molecular absorption parameters and other atmospheric effects, such as incorporating single and multiple scattering, vertical profiles of atmospheric composition, pressure and temperature, and effects of clouds, rain, and airborne particles, with the main focus in the meteorological and climatological fields.

Parallel to the model development, databases of molecular absorption data were compiled, allowing simulation and detection of variable atmospheric composition, pressure, and temperature in far greater ranges then measured in the Earth’s atmosphere. The high-resolution transmission molecular absorption database (HITRAN) [8] is a compilation of theoretically- and experimentally-derived spectral line absorption data by numerous authors, first introduced in 1973 and frequently updated since. This database was well accepted in the scientific community and considered the best source to date.

Accompanying the HITRAN database since the 2012 edition were collision-induced absorption (CIA) spectrums of the most significant binary complexes of molecules in bound, quasi-bound, and free scattered states [9]. The absorption by complexes of more than two molecules was expected to be insignificant and omitted in the database. Hartmann [10] used HITRAN line-by-line data, as well as FORTRAN simulated collision induced absorption data to isolate N_2_ absorption spectrum near 2.16 μm and to measure nitrogen content of the Earth’s atmosphere. Validation of CIA simulation was performed by fitting absorption spectrum to the difference in balloon-borne and ground-based (solar zenith angle ≥ 87°) measurements of solar radiation. Fitting residuals were shown to improve from 5% without collision-induced absorption to 2% with collision-induced absorption, whereas the measurement-fitted N_2_ content parameter 0.783 ± 0.022 successfully corresponded with the expected reference value of 0.7809. Hartmann’s N_2_ Air CIA spectrums were also included in the most recent (2016) edition of the HITRAN CIA database.

## 2. Materials and Methods

### 2.1. Calculation of Moist Air–Gas Ratios

According to Dalton’s law, partial pressure of sum of all dry gases $pp_{dry\_gases}$ is reduced proportionately when an amount of water vapour $pp_{water\_vapour}$ is added at a constant atmospheric pressure p. Assuming a constant molar composition of dry air, partial pressures of individual gases are equally reduced when water vapor is added, which should not be true for real gas components with different individual compressibility factors.
(1)$$p=pp_{dry\_gases}+pp_{water\_vapour}$$

Ideality of Dalton’s law is also assumed within the empirical formulations of CIPM-2007 [11] and Hardy ITS-90 [12], which account for nonideality of the moist mixture when calculating total compression; however, within the mixture, fractions of partial pressures and molar fractions are always considered proportional. Equal assumption of linearity was used in this model, where partial pressure shift and molar ratio shifts were considered equal at a constant temperature and pressure, disregarding a change in moist air composition with a changing vapor content.

Partial pressure of the water vapor is thus calculated according to CIPM-2007 formulations for the Celsius scale (where Celsius temperature is denoted with t to differentiate from thermodynamic temperature T) as a fraction of saturation partial pressure of water vapour, scaled by the relative humidity h, defined in the units of relative range of 0%–100% relative humidity (% RH),
(2)$$pp_{water\_vapour}=pp_{sat}\left(p,t\right)\;\frac{h}{100\%\;RH}$$
where $pp_{sat}\left(p,t\right)$ is a saturation partial pressure, calculated from a saturation partial pressure in vacuum $e_{s}\left(T\right)$, adjusted by an enhancement factor f(p,t), to account for pressure of other gases in a mixture.
(3)$$pp_{sat}\left(p,T\right)=f\left(p,t\right)\;e_{s}\left(T\right)$$

### 2.2. Dry Air Composition

Dry air composition, gathered from multiple sources is listed in Table 1. Remaining gases are of trace amounts and are neglected. Noble gases (gray in Table 1) are atomic gases, considered transparent to vibrational interaction, and not included in further algorithm. Due to rising CO_2_ levels and lack of recent measurements of remaining gas fractions, O_2_ levels were considered to decrease in proportion to CO_2_ increase. This hypothesis was previously considered in a local environment as the effect of combustion and breathing by Krogh [13], cited by Paneth [14], and accepted by Giacomo [15] for CIPM 81/91 formulation, observed by Keeling [16] as an effect in urban areas, and used by Gatley et al. [17], Ginzburg [18], and Park [19] for adjusting global levels of CO_2_ and O_2_. Schlatter [20] evaluated that 93% of O_2_ loss from the atmosphere was related to respiration and decay of animal life and bacteria, with consumption and burning of biomass fuels causing another 4% loss. Following this simplification, effect of increased solubility of CO_2_ over O_2_ in seawater, as reported by Keeling [16], was neglected. Final molar fractions used when exporting are listed in Table 2. Note that water vapor content is adjusted later in the algorithm; however, a reference molar content of 8391 ppm was used, corresponding to normal values of 30% RH at temperature 296 K.

### 2.3. Spectral Transmissivity Calculation

Transmissivity spectrum τ(λ) of homogenous mixtures is calculated from the Beer–Lambert law:(4)$$\tau \left(\lambda \right)=e^{-OD\left(\lambda ,pp,p,T\right)}=e^{-\sum^{\;}\sigma_{i}\left(\lambda ,pp,p,T\right)\;\rho_{i}\;l}=e^{-\sum^{\;}A_{c,i}\left(\lambda ,pp,p,T\right)\;l}$$
where OD(λ) represents optical density of the transfer path, which is the sum of optical densities of each gas. Optical density of each gas is calculated as a product of attenuation cross section σ(λ), number density ρ, and path length l. Alternatively, optical density can be expressed as a product of path length and absorption coefficient $A_{c}\left(\lambda \right)$.

In his paper, Karman [9] further describes absorption coefficient of a species as virial expansion in the number density:(5)$$A_{c,i}\left(\lambda ,pp,p,T\right)=\sigma^{\left(1\right)}{}_{i}\left(\lambda ,pp,p,T\right)\;\rho_{i}+\sigma^{\left(2\right)}{}_{i}\left(\lambda ,pp,p,T\right)\;\rho_{i}{}^{2}+\ldots$$
where the first cross-section coefficient $\sigma^{\left(1\right)}{}_{i}$ represents absorption by monomers, such as given in line-by-line database, the second coefficient $\sigma^{\left(2\right)}{}_{i}$ represents absorption by pairs of two molecules, and so on. The second virial coefficients of cross sections are square-dependent of pressure and appear as not of significant value under usual atmospheric conditions (Figure 1)—nevertheless, binary complexes are still represented in the final model.

Line-by-line absorption data in Figure 1 was obtained from The HITRAN2016 molecular spectroscopic database [8] and converted to absorption coefficient spectrum $A_{c}\left(\lambda ;p,pp,T\right)$ using the MATLAB function Load Hitran by DeVore [26] for set individual gas component of a calculated number density. Within the nine molecular species presented in Table 2, a total of 25 isotopologs have been accounted for. An isotopolog is a variation of a molecule where at least one of its atoms are an isotope. Isotopologs of total molar concentrations of less than 0.01 ppm in dry air were omitted from the calculation. Abundance values were sourced from the HITRAN2016 database, whereas atomic masses of isotopes were obtained from Bievre et al. [27]. Absorption cross section of species i is calculated for multiple isotopologs indexed with j and weighted corresponding to isotopolog abundance $a_{j}$.
(6)$$\sigma_{i}\left(\lambda ;\rho ,\rho_{i\_ref},T\right)=\sum^{\;}\sigma_{i,j}\left(\;a_{j}\;\rho_{i}\right)$$

The sum of abundance-weighted absorption cross sections of a molecular species is exported to numeric spectrum and multiplied by number density $\rho_{i}$ of the molecular species and path length l. Note that constant gas species number densities were used upon exportation to include abundance data of all dry gases in one file, therefore correction factor $\rho_{i}/\rho_{i\_ref}$ is used afterwards—one for dry gases with presumably constant composition, and one for water vapor.
(7)$$OD_{i}\left(\lambda ;p,pp_{2},T\right)=\sigma_{i}\left(\lambda ;\rho ,\rho_{i\_ref},T\right)\;\rho_{i\_ref}\frac{\rho_{i}}{\rho_{i\_ref}}\;l$$

Number density, referring to the number of molecules in a unit of volume, is calculated using the gas law for real gases:(8)$$\rho =\frac{N_{A}\;n}{V}=\frac{N_{A}\;p}{Z\left(x_{v},p,T\right)\;R\;T}$$
where $N_{A}$ is Avogadro’s constant, n is amount of substance, V is volume, p is air pressure, $Z\left(x_{v},p,T\right)$ is compressibility factor of air mixture (from Picard et al.’s CIPM-07 formulation [28] to compensate for its nonideal behavior),$\;x_{v}$ is molar fraction of water vapor in air, R is the gas constant, and T is the thermodynamic temperature of the gas mixture.

### 2.4. Compiling of the MATLAB Model

Compilation to numeric spectrum is possible using the Load Hitran function [26]; however, this task is highly computationally demanding, requiring up to several hours to complete at high resolution. An optimized version of this function would be desired for the purpose of repeated simulations.

The model can be computationally optimized by compiling the line-by-line data at reference parameter values, utilizing the Load Hitran function to obtain absorption coefficient spectrums of the dry air and the water vapor components, and subsequently adjusting the spectrums for the situational specific parameters. This two-part separation of calculations requires a single-time computationally demanding compilation and permits fast recall and adjustment of the saved data whenever the spectral transmissivity model is required.

It is important to point out that the described separation is not entirely theoretically correct. As temperature, partial and remaining pressure, and, thus, also water vapor concentration are influential parameters of the Load Hitran function, specifically influencing the pressure- and temperature-dependent Voigt line shape, each individual conversion of line-by-line data to numeric spectrum must theoretically be conducted at individual parameter values. While spectral line shift and broadening are theoretically present for parameter deviations, considering the relatively narrow range of expected atmospheric parameter values, the change in these effects can be minimized and neglected by setting reference parameters of the single-time compilation to expected usual values. Reference values are listed in Table 3.

Separately, collision-induced cross sections of N_2_–air binary complexes, supplied by Hartmann [10] in the HITRAN CIA database, were simulated for the dry gas mixture.

The final algorithm of the MATLAB model for transmission spectrum calculation, with influential parameters of relative humidity, air pressure, and temperature, is displayed in Figure 2. The resulting MATLAB code produces spectral transmissivity of 1–18 µm for variable relative humidity, temperature, and atmospheric pressure, as influenced by number density of molecules. The program does not account for intrinsic line broadening and shift as an effect on particles due to variations in partial pressures and temperature. Final spectral transmissivity is plotted in Figure 3.

Total transmissivity of radiation of all wavelengths is displayed in Figure 4. Considering sensor spectral sensitivity, we use two different radiation thermometers in our laboratory. KT 19.01 II (SP01) measures in the spectral range of 2.0–2.7 µm and at temperatures between 350–2000 °C, and the Transfer Radiation Thermometer (TRT II) with ranges of 8–14 µm (SP82) for temperatures between −50–300 °C, and at 3.87 µm (SP41) for temperatures between 150–1000 °C.

Spectral sensitivity characteristics, digitized from Heitronics pyrometer documentation [29] using WebPlotDigitizer [30] and imported into simulation environment, are displayed in Figure 5. Alongside these, generic vanadium oxide (VOx)-coated microbolometer spectral sensitivity by FLIR [31] was also imported, representing uncooled thermal imagers in the 8–14 µm range. Amplitude of each sensitivity curve is irrelevant, as the function fit of temperature vs. radiative power characteristic was calculated for each sensor individually.

Effects of the atmosphere at described settings are displayed in Figure 6. Transmissivities were calculated by simulating blackbody radiation, transmitted through atmosphere, and absorbed by the sensor of relative spectral sensitivity from Figure 5. Total transmissivity represents a fraction of the total radiation that is transmitted.

Measurement errors were simulated by converting radiation, influenced by atmospheric absorption and emission over inverse radiation–temperature characteristic of the sensor, compiled for the black body source.

In accordance with the Beer–Lambert law (Equation (4)), one can adjust transmissivity $\tau_{old}$ to different path length $l_{new}$ by applying exponential correction factor to $\tau_{old}$ of path length $l_{old}$.
(9)$$\tau_{new}=e^{-A_{c}\;\frac{l_{new}}{l_{old}}l_{old}}={\left(\tau_{old}\right)}^{\frac{l_{new}}{l_{old}}}$$

Similarly, number density correction can be applied.
(10)$$\tau_{new}={\left(\tau_{old}\right)}^{\frac{n_{new}}{n_{old}}\;\frac{l_{new}}{l_{old}}}$$

Considering that most pyrometers do not support transmissivity adjustments, correction must be applied manually, if necessary. Assuming equilibrium, where temperature of the environment is equal to temperature of the atmosphere in the radiation transfer path through the air, transmissivity factor can be applied to emissivity setting of the instrument using multiplication, considering that mathematical effect of these two parameters is equal. Furthermore, automatic compensation of emissivity is usually calculated using the temperature of radiation thermometer’s sensor inside the enclosure, which corresponds to the atmospheric temperature.

### 2.5. Comparison with Experimental Results

Practical influence on radiation thermometer was measured by Omejc [32], using a Vötsch VC7100 climatic chamber, where the radiation thermometer Heitronics KT19.01 II and radiation source FLUKE 4181 were positioned closely outside the chamber (Figure 7).

The manufacturer of the radiation source specifies emissivity of the flat-plate radiation thermometer calibrator at 0.95. In the experiment, the radiation transfer path included a 15 cm gap on the source side and extended through the climatic chamber openings, located 1.3 m apart. The diameter of the openings is 125 mm at the side of the radiation source and 50 mm at the other side, where the radiation thermometer was placed with neglectable gap. The experimental system was simulated using the compiled model of atmospheric spectral transmissivity by simulating two homogenous zones of gases, one within the chamber, with variable humidity and temperature of 23 °C, and one outside the chamber, between the opening and the radiation source, with heater-induced increased temperature (mean increase of 10% of total temperature difference) and ambient humidity of 30% RH. Instrumental emissivity setting was applied to the simulation to match the experimental setting. The systematical error of the radiation source from a postponed accuracy evaluation was included in the simulation; however with a time difference of over a year, possible drift error was not accounted for. Results of simulation of the experiment and measurements are presented in Figure 8.

While the errors between simulation and measurement results could indicate incorrect behavior of the model, discrepancy between the results can at least partially be attributed to experiment realization uncertainty. For example, the size of the openings can locally distort the homogeneity of the climatic chamber, increasing the uncertainty of the experiment. The temperature outside the opening of the climatic chamber is also only approximated, introducing additional uncertainty to the simulation results.

## 3. Results

Results of sensor responses suggest that, amongst the instruments in use in the calibration laboratory practices, the effect of atmospheric interference is relatively high for the reference radiation thermometer Heitronics KT19.01 II and moderate for the low-temperature sensor on the Heitronics TRT II and vanadium oxide-based thermal cameras.

The constant offset between the results can be attributed to error in radiative temperature realization, whereas the measurement error appears to correlate between the simulation and the experiment, indicating correct behavior of the simulation-predicted sensitivity of at least the KT19.01 II radiation thermometer.

## 4. Discussion

Fractions of water vapor and CO_2_ mainly contribute to the spectral transmissivity of the air. Results of simulation reveal important atmospheric effects for some sensors, which need to be accounted for when calibrating and using these instruments for measurement. It is especially important in a practical calibration procedure of a radiation thermometer to account for influential parameters at atmospherically sensitive spectral ranges and to match atmospheric conditions during calibration as close as possible to the conditions in the target application (mainly moisture levels and the measurement distance). When conditions do not match with conditions of calibration, the calculated errors should either be taken into account for correction or included into the measurement uncertainty.

## 5. Conclusions

The sensitivity of the reference radiation thermometers KT19.01 II to atmospheric influential parameters is evidently high, therefore special attention needs to be paid to this thermometer during calibration and deployment.

Existing laboratory practice of listing permissible range of atmospheric parameters on the calibration certificate, which is relatively wide, is evidently not appropriate for this thermometer, as it significantly contributes to the calibration uncertainty, which was formerly underestimated.

To ensure the correct transfer of the reference temperature between the instrument calibration and deployment of the radiation thermometer as the reference for lower tier calibration traceability, matching, monitoring, documentation, and possible correction are necessary for the main influential parameters—measuring distance, relative humidity, atmospheric temperature, and pressure.

Further experimental research of atmospheric influences and comparison with the described model will be conducted in the future, by controlling the transfer path of calibration in front of the black body with the help of a climatic chamber with increased flow and better sealing at the radiation path openings.

## Figures and Tables

**Figure 1 sensors-21-05509-f001:**
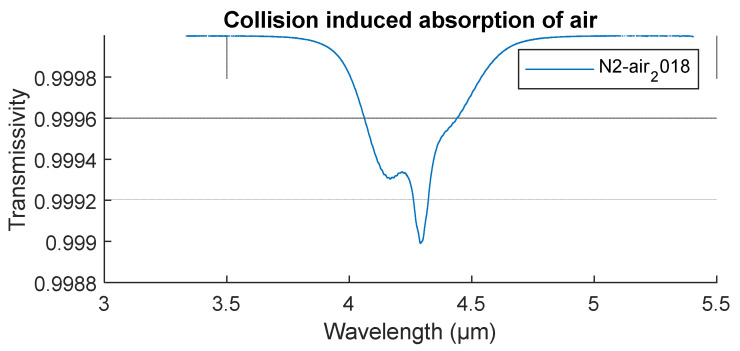
Collision-induced absorption of air.

**Figure 2 sensors-21-05509-f002:**
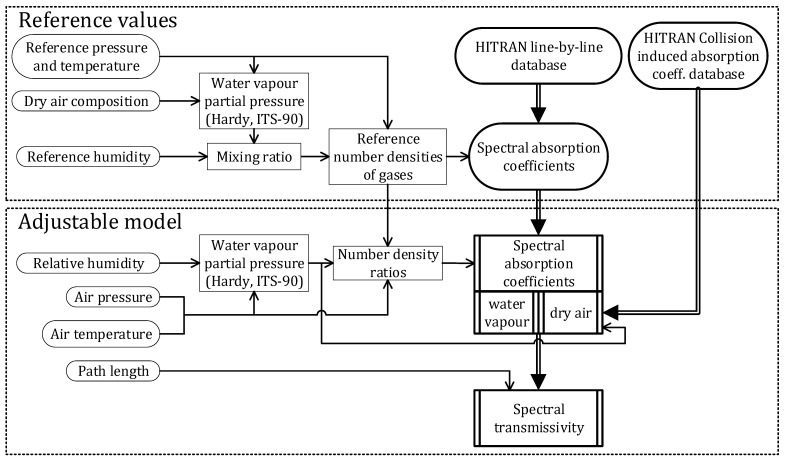
Flow chart of air transmissivity calculation algorithm.

**Figure 3 sensors-21-05509-f003:**
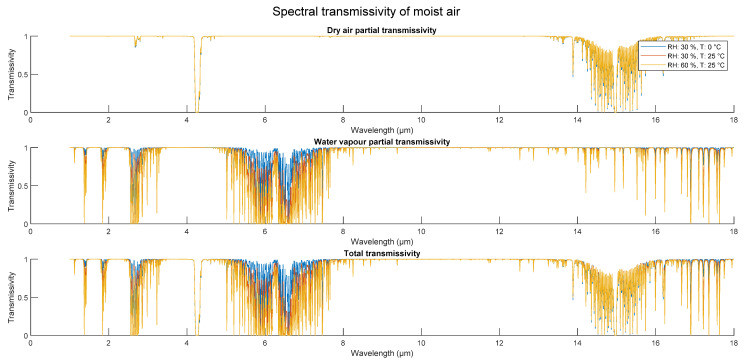
Spectral transmissivity of air at various parameters. Pressure is set to 1013.25 hPa.

**Figure 4 sensors-21-05509-f004:**
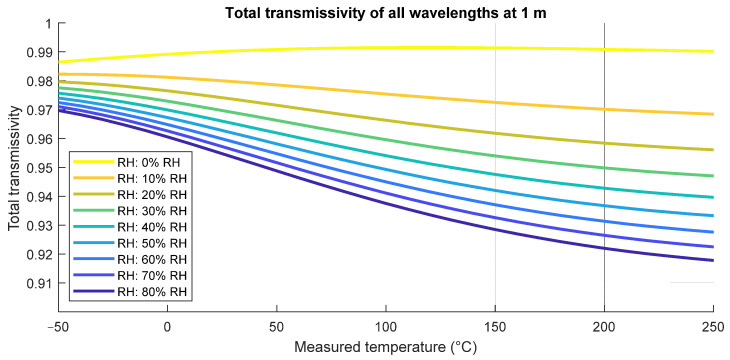
Total transmissivity of the black body radiation of wavelengths between 1 and 18 µm depending on relative humidity.

**Figure 5 sensors-21-05509-f005:**
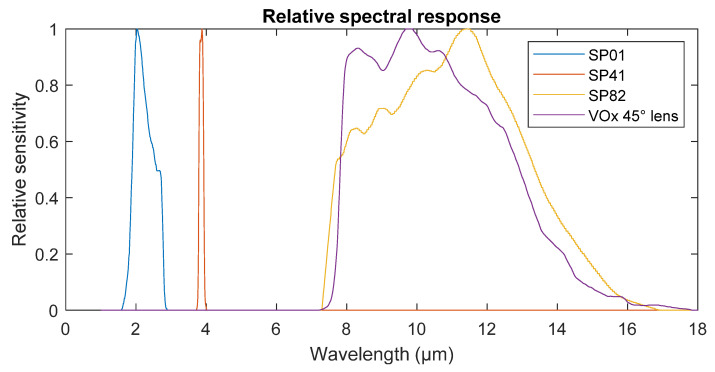
Relative spectral sensitivity of sensors.

**Figure 6 sensors-21-05509-f006:**
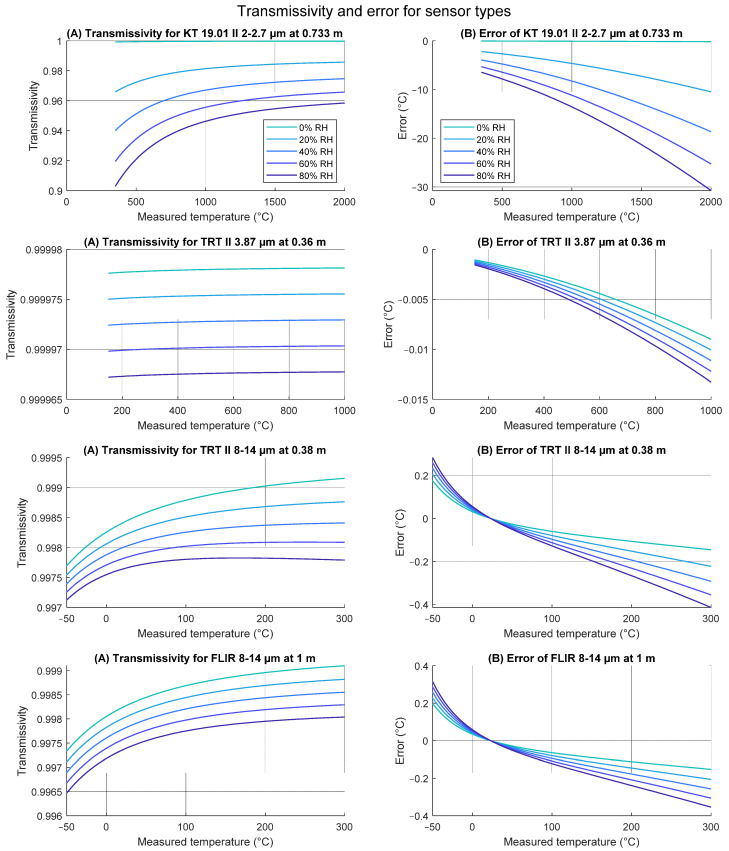
Considering the sensor spectral sensitivity, (**A**) the effective transmissivity is plotted and (**B**) the calculated error of measurement due to the atmospheric transmissivity including radiation contribution at the set focal distance, air pressure 1013.25 hPa, and temperature 296 K for various relative humidity levels is plotted. The focal distance of the thermal imaging camera is adjustable and was set to 1 m.

**Figure 7 sensors-21-05509-f007:**
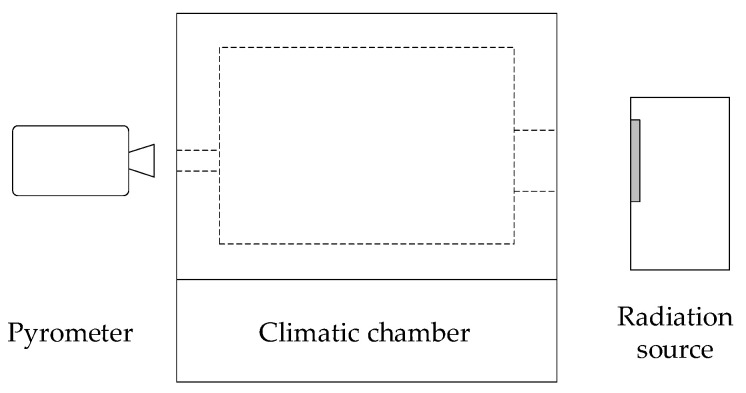
Sketch of the validation experiment.

**Figure 8 sensors-21-05509-f008:**
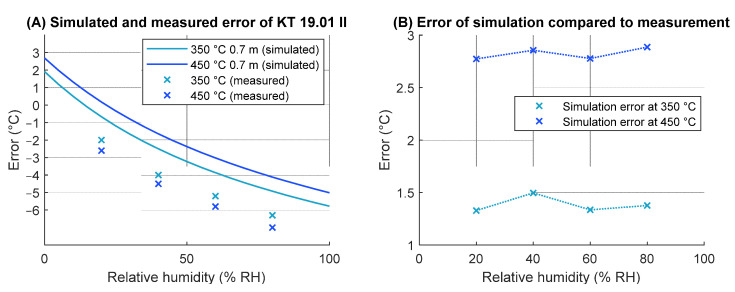
Comparison of experimental and simulated measurements. (**A**) The error represents measurement deviation from reference temperature and (**B**) the error represents discrepancy between the simulated measurement and actual measurement.

**Table 1 sensors-21-05509-t001:** Molar and volumetric fractions of gases in dry air at the sea level in ppm. In bold are the latest values, which were used for the final values.

Citation	[21]	[22]	[17]	[11]	[19]	[16]	[15]	[23]	[20]	[24], in [18]	[25]
Source	NOAA	US EPA trends	Gatley et al.,	Picard CIPM07	Park et al.	Keeling	Giacomo CIPM 81	NASA Factsheet	Schlatter	Atmosfera—Gidrometeoizdat	US Std Atm 62
Year	2020	2019	2004–8	2007	2004	1970–86	1981	2020	2009	1991	1954
Fraction	Mole	Mole	Mole	Mole	Mole	Mole	Mole	Volume	Volume	Volume	Volume
N_2_			**780,818**	780,848	781,010	780,670	781,010	780,800	780,840	780,840	780,840
O_2_			**209,435**	209,390	209,390	209,460	209,390	209,500	209,460	209,460	209,476
H_2_O			0	0	0	0	0	0	0	0	0
Ar			**9332**	9332	9170	9340	9170	9340	9340	9340	9340
CO_2_	**412.4**		385	400	400	339	400	410	384	394.45	314
Ne			**18.2**	18.2	18.2	18.18	18.2	18.18	18.18	18.18	18.18
He			**5.2**	5.2	5.2	5.24	5.2	5.24	5.24	5.24	5.24
CH4	**1.8923**		1.5	1.5	1.5	1.7	1.5	1.7	1.774	1.79	2
Kr			**1.1**	1.1	1.1	1.14	1.1	1.14	1.14	1.14	1.14
H2			**0.5**	0.5	0.5	0.5	0.5	0.55	0.56	0.55	0.5
N2O	**0.3336**		0.3	0.3	0.3	0.3	0.3	-	0.32	0.325	0.5
CO		**1.08**	0.2	0.2	0.2	0.025	0.2	-	-	0.1	-
Xe			**0.1**	0.1	0.1		0.1		0.09	0.09	0.087
O3		**0.064**				0 to 0.1			0.01 to 0.1	0 to 0.07	0.02 to 0.07

**Table 2 sensors-21-05509-t002:** Gas molar fractions of compounds used in this study. Reference content settings are values used when exporting from HITRAN data to numeric spectra.

Compound	Name	Content in Dry Air (ppm)	Reference Content $x_{i}(\mathrm{ppm})$
N_2_	Nitrogen	780,818	774,352
O_2_	Oxygen	209,407.6	207,673
H_2_O	Water	0	8391
Ar	Argon	9332	
CO_2_	Carbon Dioxide	412.4	408.98
Ne	Neon	18.2	
He	Helium	5.2	
CH_4_	Methane	1.8923	1.876
Kr	Krypton	1.1	
H_2_	Hydrogen	0.5	0.496
N_2_O	Nitrous Oxide	0.3336	0.331
CO	Carbon Monoxide	1.08	1.071
Xe	Xenon	0.1	
O_3_	Ozone	0.064	0.063

**Table 3 sensors-21-05509-t003:** Parameter set-point during spectrum preparation.

Symbol	Value	Quantity
RH	30%	Relative humidity of air
T	296 K	Air temperature
$p_{ref}$	1013.25 hPa	Air pressure
$\rho_{ref}$	2.48∙10^19^ cm^−3^	Number density of air *
$\rho_{i\_ref}$	$x_{i}\cdot \rho_{ref}$	Number density of a molecular species
dλ	0.1 nm	Spectral resolution
l	10 m	Path length (used only in plots)

* value preset in Load Hitran code and corrected later in the algorithm.

## Data Availability

The data presented in this study are available on request from the corresponding author. The data are not publicly available due to size restrictions.

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
