# Peer review of "Influence of Atmosphere on Calibration of Radiation Thermometers"

_sensors, 2021, doi:10.3390/s21165509_

Round 1

Reviewer 1 Report

What is called "isotopologues" in the Abstract? I have not found this word in the text of the article.

Author Response

Isotopologue values are used in equation 6 and briefly mentioned in the preceding paragraph. A short description has now been added to the beginning of the paragraph to improve the presentation.

An isotope is a term, describing atoms of the same species (defined with proton count) with different numbers of neutrons. Isotopologue is a chemical, differing from its parent chemical in that at least one of its atoms is an isotope. In particular, following table displays all the isotopologues used in this paper. Isotopologues of total molar concentration of less than 0.01 ppm in dry air were neglected and omitted in the study.

Species

Isotopologue

Abundance of isotopologue (relative to species)

Atomic masses

N2

14N14N

0.99268700

28.00614802

15N14N

0.00747800

29.00318299

O2

16O2

0.99526200

31.98982928

16O18O

0.00399100

33.99407403

16O17O

0.00074224

32.99404514

CO2

12C16O2

0.98420400

43.98982928

13C16O2

0.01105700

44.99318412

16O12C18O

0.00394700

45.99407403

16O12C17O

0.00073399

44.99404514

16O13C18O

0.00004434

46.99742887

CH4

12CH4

0.98827400

16.03130015

13CH4

0.01110300

17.03465499

12CH3D

0.00061575

17.03757690

H2

H2

0.99968800

2.015650074

HD

0.00031143

3.021926824

N2O

14N216O

0.99033300

44.00106266

CO

12C16O

0.98654400

27.99491464

O3

16O3

0.99290100

47.98474392

H2O

H216O

0.99731700

18.01056471

H218O

0.00200000

20.01480946

H217O

0.00037188

19.01478057

HD16O

0.00031069

19.01684146

HD18O

0.00000062

21.02108621

HD17O

0.00000012

20.02105732

D216O

0.00000002

20.02311821

In the attached image is an example of the transmissivity spectrum of all isotopologues, used for the Figure 3. Note that this image is not from the final stage of development, however the values of concentrations are close to correct.

Reviewer 2 Report

The authors analyze in this work the influence of various compositions in atmospheric air to the transmission spectrum in the mid infrared range, with the aiming to provide a reference for calibration in a practical measurement. I have the following comments for the authors to consider.

  1. The tilte "Influence of Atmosphere in Calibration of Radiation Thermometers" is misleading. The emphasis of this work is to analyze the influence, which can potentially provide the reference for calibratio. However, no calibrationn is conducted at all in this manuscript. An example of the calibration should be provided if the authors continue to use this title to show.
  2. All the results presented in this work can be numerically obtained using the HITRAN database. The authors claim that the calibrations of radiation thermometers are being conducted for more than 20 years in their institution. So some experimental work should be at hand and is recommended to include in this manuscript for comparison.

Author Response

Thank you for your input.

  1. Indeed, the manuscript is intended to analyze the influence on the calibration of radiation thermometers. I have edited the title accordingly.
  2. A chapter on comparison between the simulated and measured results of an experiment using a climatic chamber has been added near the end of the manuscript.

Reviewer 3 Report

Basically, the article deals with an important and interesting issue of the influence of water vapor content on air transmissivity.
The downside of the work is the simplified and cursory presentation of the results and the way of comparing the measurements with the reference values. A more extensive description of the measurements and comparisons would increase the value of the article. The current version of the article makes it less attractive than it could be.  

There is no order in the article.
Referring to the literature data (Omejc) in the conclusions is too late. This is not a good place for this type of information.

The second drawback of the article is its careless editing. Description is missing for some variables. In equation (1) it must be assumed that p is the total pressure, but the partial pressure pp is described.

In equation (2) and (3)  ppsat(p,t) and f (p, t) appears. What does t mean? 

On line 119 there is a double reference to Table 1.

The article in the part concerning measurements (probably beginning from line 204) should be redrafted and corrected.

Author Response

Thank you for your input.

I have added a section to clearly compare the results of the experiment (by Omejc) and added a simulation.

Text has been separated into chapters to more clearly present the purpose of each chapter.

Thank you for pointing out the missing definitions of variables. I have edited definitions throughout the manuscript.
As now also noted in the text, t denoted a temperature in the Celsius scale.

A chapter on comparison with experimental measurements has been added, and the remainder of the text was redrafted, according to the instructions.

Reviewer 4 Report

In the manuscript entitled “Influence of Atmosphere in Calibration of Radiation Thermometers”, the authors have carried out an analysis of relative humidity, temperature, air pressure and path length on the thermal radiation transmissivity. Though results show relatively low changes, as can be seen from Figures 4 and 6, this work could be of interest for researchers in the area to make further improvements in the thermal sensor calibration processes. However, before I recommend publication of this manuscript, I recommend that the authors improve the quality of the manuscript. In particular, the introduction section of a paper is not the same style of an introduction of a project. The authors should work on the introduction section to improve its quality and presentation.

Manuscript presentation can also be confusing. Throughout the manuscript the authors discuss measurements and corrections in relation to their own thermometer, while the results are only discussed for numerical simulations (using available data sets). The authors should clearly emphasize if this work is about numerical simulations or experimental work. Although simulations could (evidently) be of interest for future experimental work, the readers need to unambiguously know the kind of contribution from this work.

Some minor points follow:

  • Line 41, the authors should define RH. They used it without definition in line 133.
  • Line 119, Table 1 is repeated.
  • The idea between lines 179 and 180 is not clear.
  • Line 186. I recommend changing the word spectre to spectrum.

Author Response

Thank you for your input. Indeed, the influence is relatively low, but of significant contribution to the overall measurement uncertainty.

I have revised the introduction section to present the purpose of the research more clearly.

I would like to point out that most used datasets do indeed correspond and are traceable to the actual equipment in use, therefore the results of the numerical simulations are considered to also correspond to the actual instruments. To clarify,

  • HITRAN data originates from numerous scientific papers and is most importantly traceable to experiments, despite having disregarded the uncertainty data, which is included in the database for each individual line.
  • Heitronics spectral sensitivity data are actual measurements of the sensor properties, provided by the manufacturer, and are therefore complementary to the instruments in use. However, unknown uncertainty and further uncertainty after graphical visualization and backward digitalization of the data do indeed indicate the need for validation. Opposingly, Vanadium Oxide spectral response is generic and reportedly not traceable to actual instruments, therefore the results for thermal cameras are not to be trusted, however do provide a good assessment on instrument’s low susceptibility in the corresponding spectral range of 8 µm—14 µm.

Considering this, I have revised the abstract and introduction chapter to match the purpose of the research, presented in the manuscript.

Round 2

Reviewer 3 Report

The article has been significantly improved. Reading is easier, especially thanks to the changes in chapter 2.5. Comparison with experimental results and 2.4. Compiling of the MATLAB model

There is still an error on line 140: Table 1Table 1

Reviewer 4 Report

I want to thank the authors for taking the time to revise all my concerns, which were properly addressed. I recommend publication of the manuscript in its present form.